# A Phase II, Multicenter, Single-Arm Study of Mipsagargin (G-202) as a Second-Line Therapy Following Sorafenib for Adult Patients with Progressive Advanced Hepatocellular Carcinoma

**DOI:** 10.3390/cancers11060833

**Published:** 2019-06-17

**Authors:** Devalingam Mahalingam, Julio Peguero, Putao Cen, Sukeshi P. Arora, John Sarantopoulos, Julie Rowe, Victoria Allgood, Benjamin Tubb, Luis Campos

**Affiliations:** 1Department of Medicine, Division of Hematology/Oncology, Northwestern University, Chicago, IL 60611, USA; 2Mays Cancer Center, University of Texas Health Science Center, San Antonio, TX 78229, USA; aroras@uthscsa.edu (S.P.A.); sarantopoulo@uthscsa.edu (J.S.); 3Oncology Consultants Research, Houston, TX 77030, USA; jpeguero@oncologyconsultants.com (J.P.); lcampos@oncologyconsultants.com (L.C.); 4Internal Medicine, University of Texas Health Science Center, Houston, TX 77030, USA; putao.cen@uth.tmc.edu (P.C.); julie.rowe@uth.tmc.edu (J.R.); 5GenSpera, Inc. San Antonio, TX 78258, USA; victoria.allgood@gmail.com; 6Intrinsic Imaging, LLC, San Antonio, TX 78212, USA; btubb1@gmail.com

**Keywords:** Mipsagargin, G-202, prodrug, hepatocellular carcinoma

## Abstract

*Background*: Mipsagargin (G-202) is a thapsigargin-based prodrug with cytotoxic activity masked by a peptide that is cleaved by prostate-specific membrane antigen (PSMA), a protease expressed in prostate cancer cells and the endothelium of tumor vasculature. It was hypothesized that PSMA-mediated activation of mipsagargin would result in disruption of the tumor vasculature, leading to a decrease in blood flow, and in direct cytotoxic effects on tumor cells, resulting in anti-tumor activity. *Method*: In this open-label, Phase II study, mipsagargin was administered intravenously on Days 1, 2, and 3 of a 28-day cycle to patients with hepatocellular carcinoma (HCC) who progressed on or after treatment with sorafenib or intolerant of sorafenib. Assessments included time to disease progression (TTP), response rate, progression-free survival (PFS), overall survival (OS), and safety. Blood flow metrics in hepatic lesions were evaluated using dynamic contrast-enhanced magnetic resonance imaging (DCE-MRI). *Results*: Of 25 treated patients, 19 were evaluable for efficacy. None had an objective response, 12 (63.2%) had a best response of stable disease, and 12 (63.2%) showed radiologic progression; seven patients (36.8%) were censored. The median TTP was 134.0 days, median PFS was 129.0 days, and median OS was 205.0 days. Of five patients with DCE-MRI data for 11 HCC lesions, all demonstrated a reduced K_trans_ (mean, 52%). The most common treatment-emergent AEs were Grade 1–2 and consisted of increased blood creatinine (68.0%), fatigue (56.0%), and nausea (44.0%). *Conclusions*: Mipsagargin is relatively well tolerated and promotes prolonged disease stabilization in patients with advanced HCC that had progressed on prior treatment with sorafenib. A significant decrease in K_trans_ upon treatment suggests mipsagargin reduces blood flow in hepatic lesions.

## 1. Introduction

Worldwide, liver cancer is the second most common cause of cancer death in men and the sixth most common cause in women due to its poor prognosis, and hepatocellular carcinoma (HCC) accounts for 70% to 90% of primary liver cancers [1]. The median survival time is 16 to 20 months for those presenting with an intermediate stage of disease, and only six months from diagnosis for patients presenting with unresectable, advanced-stage disease [2]. Unfortunately, most HCC cases are diagnosed at an advanced stage, when surgical resection is only suitable for approximately 5% of patients. For the last decade, the kinase inhibitor sorafenib (Nexavar^®^, Bayer Healthcare, Whippany, NJ, USA) was the only approved treatment for unresectable HCC, although the efficacy for this agent is modest, with a median survival of less than 11 months [2,3,4]. Despite the evolution of systemic therapy options, many patients with advanced HCC still succumb to this disease [5].

HCC is a highly vascularized tumor [3], and ongoing angiogenesis may be essential to its growth, invasion, and metastasis [6,7,8]. Angiogenic factors such as angiopoietin, vascular endothelial growth factor (VEGF), platelet-derived growth factor (PDGF), and fibroblast growth factor-2, inflammatory cells, and/or tumor stromal cells participate in the neovascularization of HCC [9,10,11]. The enhanced vascularity of HCC allows for radiological diagnosis of HCC. This high degree of vascularization may make HCC particularly sensitive to therapies targeting the tumor neovasculature.

The prostate-specific membrane antigen (PSMA) is a 110-kDa cell surface transmembrane glycoprotein. It functions as a glutamate carboxypeptidase to hydrolyze the γ-glutamyl linkages of glutamate chains of various lengths [12,13,14]. PSMA is weakly expressed in normal prostate and is strongly upregulated in prostate cancer; it is also expressed on the apical surface of endothelial cells within the tumor vasculature of many tumor types, but it is not expressed in the vasculature of normal tissues or the epithelium of most normal tissues [15,16,17,18,19]. One recent study reported that in 40 of 42 (95%) HCC tumor tissue specimens, PSMA was detectable by immunohistochemical staining; in contrast, PSMA expression was not detected in the vasculature of a normal liver (*n* = 9 cases) [18,19]. Because PSMA is expressed in the vasculature of HCC tumor tissue but not in a normal liver, PSMA-directed targeting of anti-cancer agents might be a particularly effective approach for HCC.

Mipsagargin (G-202; chemical name (8-O-(12-aminododecanoyl)-8-O-debutanoylthapsigargin) aspartate-γ-glutamate-γ-glutamate-γ-glutamate-glutamateOH) is a prodrug consisting of a PSMA-selective five-amino-acid-masking peptide substrate coupled to a cytotoxic analog of thapsigargin [20,21]. It was developed on the hypothesis that the extracellular enzymatic activity of PSMA could be used to effectively and efficiently cleave the masking peptide, thereby delivering cytotoxic agents specifically to a tumor site. Thapsigargin functions by potently inhibiting a critical intracellular protein, the sarcoplasmic/endoplasmic reticulum calcium ATPase (SERCA) pump, whose normal function is the maintenance of intracellular calcium homeostasis and promotion of cell viability [22,23]. Inhibition of the SERCA pump results in dysregulation of intracellular calcium levels and subsequent induction of apoptotic cell death [22,24]. When the masking peptide component of mipsagargin is cleaved by PSMA, it liberates 12ADT-Asp, a cytotoxic analog of thapsigargin [25]. Subsequently, 12ADT-Asp binds to the SERCA pump, producing a sustained elevation in intracellular calcium, which results in the activation of apoptosis [21]. Thus, mipsagargin is a first-in-class PSMA-targeted prodrug. In an early clinical study, it was shown that in addition to a good safety profile, patients with advanced HCC treated in that study had prolonged disease stabilization [26] supporting the hypothesis that a PSMA-targeted agent might indeed be particularly effective in HCC. Thus, this Phase II multicenter, single-arm study was undertaken to evaluate the safety and efficacy of mipsagargin in adult patients with advanced HCC who had progressed on or after treatment with sorafenib or were intolerant of sorafenib. 

## 2. Patients and Methods

### 2.1. Study Design and Patient Selection

This was a five-center, open-label, phase II study consisting of an initial safety run-in phase followed by an expansion phase. To be eligible, patients had to be adults with histologically, cytologically or radiologically confirmed HCC whose cancer had progressed on or after sorafenib treatment or who were intolerant of sorafenib therapy; have at least one measurable target lesion in the liver, an Eastern Cooperative Oncology Group (ECOG) performance status score of 0 or 1; a Child-Pugh score of A or B7; and have adequate and stable hematologic, renal, and liver function, an acceptable coagulation profile, and a left ventricular ejection fraction (LVEF) ≥ 50%. Patients were asked to undergo dynamic contrast-enhanced magnetic resonance imaging (DCE-MRI) before and at one timepoint after treatment for the assessment of blood flow metrics.

Patients were excluded from the study if they had undergone major surgery or received prior locoregional therapies or radiotherapy ≤ 4 weeks before the first dose of mipsagargin or had not recovered from treatment-related toxicities; were intolerant of computed tomography (CT) or magnetic resonance imaging (MRI) contrast agents; were candidates for liver transplantation; or had any other condition that, in the opinion of the investigator, would have interfered with interpretation of study results. The study was supported by GenSpera Inc., and registered at Clinicaltrials.gov (ID: NCT01777594).

### 2.2. Ethics Approval and Consent to Participate

This clinical trial adhered to the principles outlined in the Helsinki declaration and was conducted in compliance with all applicable international and national laws and regulations. The protocol was approved by the prevailing IRB at each participating institution (HSC20130231H, on 26th March 2013). All patients provided written informed consent.

All clinical data materials are maintained by clinical study sites. The study sponsor has a complete clinical data summary.

### 2.3. Treatment

Patients were pretreated with standard medications (e.g., dexamethasone, long-acting antihistamine, oral H_2_ blockers, acetaminophen) 30 min before treatment with the study drug to prevent infusion-related reactions that had previously been reported [26]. Mipsagargin was administered as an intravenous (IV) infusion over a 1-h period on Days 1–3 of each 28-day cycle and patients were allowed to continue treatment until documentation of disease progression or unacceptable toxicity. Two treatment regimens were evaluated during the safety run-in phase: mipsagargin at 40 mg/m^2^ on Days 1, 2, and 3 (Dose Level -1) and mipsagargin at 40 mg/m^2^ on Day 1 and 66.8 mg/m^2^ on Days 2 and 3 (Dose Level 1).

To maintain hydration in this patient population, an additional 250 to 500 mL of normal saline was administered by IV infusion over 1–2 h after the completion of each study drug infusion. During therapy, dose level adjustments were permitted at the Investigator’s discretion.

### 2.4. Assessments

#### 2.4.1. Antitumor Activity

The primary study endpoint was time to disease progression (TTP), with secondary endpoints of response rate, progression-free survival (PFS) and overall survival (OS). Disease status was assessed radiologically after each second cycle of treatment. Identification of target and non-target lesions and assessment of response and disease status were conducted according to the recommendations specified in the RECIST or mRECIST for HCC guidelines [27]. A secondary endpoint of the study was evaluation of the effects of mipsagargin on vascular structure, function, permeability and blood flow in HCC lesions, which were assessed in consenting patients by DCE-MRI. DCE-MRI measurements were made on a 1.5 Tesla MRI following standardized guidelines across participating sites. The volume transfer coefficient, K_trans_, was calculated using the arterial input function derived from the signal in the abdominal aorta, using a standard Tofts model for K_trans_ calculation.

#### 2.4.2. Pharmacodynamics (PD)

Several pharmacodynamic assessments were conducted. Change from baseline in serum alpha-fetoprotein (AFP) levels was evaluated before initiation of treatment and on Day 1 of each treatment cycle. For patients who had available baseline tumor tissue (archival or newly collected), tissue was collected, fixed in formalin and embedded in paraffin (FFPE) for assessment of baseline PSMA expression by immunohistochemistry (IHC) and correlated with the clinical response. 

#### 2.4.3. Pharmacokinetics (PK)

Blood samples for PK assessment were collected on Cycle 1 Day 1 pre-dose and at the end of infusion, Cycle 1 Day 2 pre-dose (approximately 24 h after the Day 1 dose), Cycle 1 Day 3 pre-dose (approximately 48 h after the Day 1 dose), and Cycle 2 Day 1 pre-dose.

#### 2.4.4. Safety

Safety endpoints included the incidence and severity of adverse events (AEs) and serious AEs (SAEs) and relationship to study drug; rate of patient discontinuation from the study due to AEs; and on-study deaths. The severity of adverse events was assessed using Common Terminology Criteria for Adverse Events (CTCAE) version 4.03 (CTEP, Bethesda, MD, USA). 

Other safety assessments included clinical laboratory parameters (hematology, serum chemistry, serum magnesium and phosphorus, coagulation parameters, and urinalysis), vital sign measurements, evaluation of LVEF and physical examinations. All safety endpoints were summarized using descriptive statistics.

### 2.5. Sample Size and Statistical Analysis

Based on a single sample of size N, an exponential survival model of time to tumor progression, 12 months of enrollment period and 12 months of follow-up, a historical median time to tumor progression of 2.1 months [28,29,30], a hypothesized median time to tumor progression of 4.2 months, two-sided testing, and a significance level of 5%, the study was designed to attain a power of 80% with *n* = 17 patients (PASS Version 08.0.8, NCSS, Kayesville, UT, USA, 2008). Assuming 25% of patients would not be evaluable for response, the required sample size for the expansion cohort of the study was up to *n* = 23. 

Efficacy analyses were planned to be conducted on the efficacy evaluable (EE) population, defined as those patients who received at least 2 cycles of mipsagargin or patients with at least one dose of study drug who discontinued therapy due to toxicity or disease progression and underwent at least one follow-up objective disease assessment. For the primary efficacy endpoint, the Kaplan–Meier product limit method was used to estimate the median time to tumor progression (TTP) in the EE population. TTP was defined as the duration of time from the first dose of mipsagargin to the time of radiologic progression. Estimated median TTP was compared with the historical value. This analysis was repeated using the ITT population for sensitivity analysis. For secondary efficacy endpoints, response rates were summarized using the number and percentage of patients with a response, along with two-sided 95% confidence intervals for the proportions. The progression-free survival (PFS) and overall survival (OS) were estimated using Kaplan–Meier product limit estimates. PFS was defined as the duration of time from the date of the first dose of mipsagargin to the date of radiologic progression or death, whichever occurred first. OS defined as the duration of time from the date of the first dose of mipsagargin to the date of death from any cause.

## 3. Results

### 3.1. Patients

In total, 25 patients were enrolled and received at least one dose of mipsagargin. The median age was 64; 18 patients (72%) were male, consistent with the preponderance of the disease in males; the majority of patients (16 patients, 64%) were white, non-hispanic. Etiology was related to Hepatitis C cirrhosis in 16 patients, with the remaining related to non-hepatitis cirrhosis. In the safety run-in portion of the study, the first three patients were treated with mipsagargin at 40 mg/m^2^ on each of Days 1, 2 and 3 (Dose Level -1). No safety concerns were observed with this regimen and the regimen was increased to 40 mg/m^2^ on Day 1 and 66.8 mg/m^2^ on Days 2 and 3 (Dose Level 1). Six patients were treated at Dose Level 1 and 1 of these 6 patients experienced a predefined dose limiting toxicity (DLT). The conservative decision was taken to enroll the remaining 16 patients in an expansion cohort under Dose Level -1. All 25 patients were evaluated for safety (safety population). Nineteen patients received at least two cycles of treatment and underwent a post-treatment radiologic disease assessment, thereby composing the efficacy evaluable (EE) population. The EE population included three patients from the initial safety run-in treated at Dose Level -1, four of six patients in the safety run-in treated at Dose Level 1 and 12 patients in the expansion cohort treated at Dose Level -1. The median time from initial diagnosis to informed consent date was 12.62 months (−0.2 to 42.0 months). Most patients (20 patients, 80%) had histologically confirmed disease. Sixteen patients (64%) had hepatitis-related etiology and the remaining were deemed to be either alcohol or non-alcoholic-steatohepatitis-related. The majority of metastatic disease sites were lung (9 patients, 36%), bone (6 patients, 24%), and lymph nodes (5 patients, 20%). All patients had been treated previously with sorafenib and no patient had a clinical response to previous sorafenib treatment, 15 patients (60%) had stable disease (SD) and eight patients (32%) had progressive disease (PD) as their best response to previous sorafenib therapy. Response to sorafenib was unknown in two patients. Other prior treatments included doxorubicin (3 patients, 12%) and 3 patients (12%) had received other chemotherapeutic regimens. Patient characteristics are presented in Table 1. 

### 3.2. Treatment Exposure

The mean duration of exposure was 10.82 (0.4 to 44.0) weeks and the overall median duration was 4.86 (0.4 to 44.0) weeks. The mean total number of infusions was 9.4 (3 to 27). The mean total cumulative dose was 702.78 (226.8 to 2115.6) mg. Overall, the mean compliance rate was 96.11%. In total, six patients had drug or dose adjustments or modifications: three patients (12.0%) had a drug or dose adjustment >20%; two (8.0%) had an adjustment of 10% to 20%; one (4.0%) had an adjustment of <10%. The overall mean percentage of drug or dose adjustment/modification by individual cycles ranged from 9.52% in Cycle 2 (*n* = 21) to 22.23% in Cycle 6 (*n* = 3). 

### 3.3. Efficacy

For analysis of the primary endpoint of TTP, 12 of the 19 EE patients (63.2%) had PD during the study. Data were censored for seven patients (36.8%). When censored data were included, Kaplan–Meier estimates showed an overall median TTP of 134.0 days (95% CI (confidence interval): 58.0, NE), with a range of 50 to 421 days (Figure 1A). The CIs were determined by the Brookmeyer and Crowley method (non-parametric), and the p-value comparison was performed using an approximate normal distribution for the Z-statistics test. The median observed TTP of 134.0 days was significantly greater than the historic control of approximately 63 days (*p* < 0.001) [28,29,30]. Sensitivity analysis in the safety population supported these findings with a median TTP of 156.0 days (95% CI: 63.0, NE), with a range of 16 to 421 days. 

Analysis of PFS was conducted using data from 17 EE patients (89.5%) who either died or had disease progression, with data censored for two patients (10.5%) (Figure 1B). When censored data were included, Kaplan–Meier estimates showed an overall median PFS of 129.0 days (95% CI: 58.0, 205.0), with a range of 50 to 421 days. Sensitivity analysis using the safety population supported these findings, with a median PFS of 113.0 days (95% CI: 58.0, 156.0) and a range of 16 to 421 days.

OS was analyzed in the EE population based on the 11 (57.9%) patients who died; the remaining eight patients (42.1%) were alive at the time of study closure and thus were censored. Kaplan–Meier estimates including censored data showed an overall median OS of 205.0 days (95% CI: 138.0, NE), with a range of 74 to 421 days (Figure 1C). Sensitivity analysis in the safety population supported these findings with a median OS of 189.0 days (95% CI: 110.0, 231.0) and a range of 16 to 421 days. 

No study patients had a complete (CR) or partial response (PR). Among the 19 EE patients, 12 were assessed as having a best response of SD, representing a disease control rate of 63.2% (95% CI: 0.4, 0.8). One additional patient, in the safety run-in portion of the study and treated at Dose Level 1 (40 mg/m^2^ on Day 1 and 66.8 mg/m^2^ on Days 2 and 3) was documented to have SD at the end of Cycle 2 before being discontinued from the study due to disease-related adverse events. The change in target lesion measurements, presented as the best response of the sum of the longest target lesion diameters, were available for 18 of 19 EE patients (Figure 2). One patient exhibited new metastatic disease and was removed from the study with an assessment of PD, so no post-treatment target lesion measurements were recorded for this patient. Among the 25 patients in the intention to treat (ITT) population, 13 of 25 patients had SD as the best response, for an overall disease control rate in the safety population of 52.0% (95% CI: 0.4, 0.8), confirming the observation made in the EE population.

### 3.4. Pharmacodynamic Assessments

No trends were detected in the changes in AFP concentration. In the absence of clinical responses, no descriptive correlation with response was performed. Archival tissue specimens were available for 13 patients; PSMA expression was assessed by immunohistochemistry in a central laboratory. In this group of patients, no clear relationship or correlation was detected between PSMA expression and any clinical outcome, safety observation, or other study parameter. 

Seven of the 19 EE patients underwent DCE-MRI assessment of tumor blood flow metrics at baseline and within three days of completion of mipsagargin administration in Cycle 2. Results from two patients were not assessable because there were no arterially-enhancing lesions in the imaging field. Among the remaining five patients, 11 HCC lesions, including 1 bone metastasis and 1 metastatic lymph node, were available for evaluation of subjective and objective evidence of decreased arterial phase hyper-enhancement and quantitation of K_trans_. All 11 lesions demonstrated a reduction in K_trans_, with an average 52% reduction (range 13–90%). Representative results are presented in Figure 3.

### 3.5. Pharmacokinetics

A bi-exponential function (without weighting) was considered to best represent the plasma concentration–time profiles of mipsagargin following a single, 1 h intravenous infusion at 40 mg/m^2^. The model adequately fit the pooled plasma concentration–time profiles of mipsagargin derived from a previous Phase I dose escalation study in patients with solid tumors treated with mipsagargin at dose levels ranging from 1.2 to 88 mg/m^2^ [26]. The parameters of this PK model were used to simulate plasma profiles of mipsagargin after three daily doses at 40 mg/m^2^ and after a single dose of 40 mg/m^2^ followed by 66.8 mg/m^2^ on two successive days. Further, plasma concentrations of mipsagargin measured at the end of infusion on Day 1 and predose on Day 2 and Day 3 in this study were within the range of plasma concentrations predicted from the previous study.

### 3.6. Safety

All 25 patients reported at least one AE which was judged by the investigator to be related to treatment with mipsagargin. Fourteen patients (56.0%) reported SAEs, of which five (20.0%) had SAEs that were related to treatment with mipsagargin. One patient in the six-patient cohort treated at Dose Level 1 (40 mg/m^2^ on Day 1 and 66.8 mg/m^2^ on Days 2 and 3) in the safety run-in portion of the study had increased Grade 3 blood creatinine, Grade 4 thrombocytopenia, and Grade 3 renal injury, which met the criteria for DLT. While there was not a second DLT in this cohort level, the dose Level 0 (40 mg/m^2^ on Days 1, 2 and 3) was conservatively selected for the expansion cohort. Five patients (20.0%) had AEs which lead to the discontinuation of treatment. No AEs resulted in death. 

All patients had CTCAE Grade 1 AEs (*n* = 25); 22 patients (88.0%) had Grade 2 AEs, and 19 patients (76.0%) had Grade 3 AEs. The most common treatment-emergent AEs were increased blood creatinine (17 patients (68.0%)), fatigue (14 patients (56.0%)), and nausea (11 patients (44.0%)). Three patients (12.0%) were reported to have had a Grade 4 AE (thrombocytopenia, increased blood bilirubin, gastrointestinal haemorrhage). There was no patient with a Grade 5 AE. The most common AEs denoted by the preferred term reported by at least 20% of patients are summarized in Table 2. 

## 4. Discussion

Systemic cytotoxic anti-cancer drugs must be able to differentiate between cancer and normal cells to effectively target tumor cells and minimize systemic toxicity. One approach for improving the safety and tumor-specific selectivity of anti-cancer agents is the use of less toxic prodrug forms that can be selectively activated in tumor tissue.

The prodrug mipsagargin was designed to selectively target the vasculature and disrupt blood flow to solid tumors. Its biological activity is based on that of thapsigargin, which induces a rapid and pronounced increase in the concentration of cytosolic calcium by blockade of the SERCA calcium pump to which it binds with high affinity [21,22,31,32,33,34,35]. However, the SERCA calcium pump is expressed ubiquitously in most tissues, making it necessary to create a delivery mechanism to release the cytotoxin preferentially at the sites of cancer or the tumor vasculature.

PSMA is highly expressed by a large proportion of primary and metastatic prostate cancers [15,17,36,37,38]. PSMA variants have also been described based on mRNA studies, including a cytosolic version of PSMA, PSM, that results from alternative splicing. Two additional alternatively spliced variants of PSMA, PSM-C and PSM-D, have also been described [39,40]. Importantly, PSMA is also expressed in the newly formed blood vessels of many tumor types, including HCC, renal, bladder, colon, neuroendocrine, pancreatic and lung cancers and the majority of breast cancers and sarcomas, but is not expressed by blood vessels in normal tissue [18,19,38]. These findings suggest PSMA-targeted prodrugs may have an application in attacking the blood supply of a large number of different tumor types, and thus have potential as a broad anti-cancer agent.

We hypothesized that the specificity and localization of PSMA expression could be used to restrict and direct delivery of thapsigargin to solid tumors, particularly the neovasculature of solid tumors. Mipsagargin is produced by coupling 12 aminododecanoyl thapsigargin (12ADT) to the beta carboxyl unit of Asp at the N-terminal end of the PSMA-substrate masking peptide Asp-γ-Glu-γ-Glu-γ-Glu-Glu to produce the prodrug 12ADT-Asp-γ-Glu-γ-Glu-γ-Glu-Glu. The Glu residues of this peptide are sequentially hydrolyzed by PSMA to release the molecule 12ADT-Asp. By linking this peptide to mipsagargin, the cytotoxic activity is masked until the peptide is hydrolyzed, at which point apoptosis of tumor-associated endothelial cells can occur, leading to the disruption of blood flow, prevention of tumor growth and subsequent tumor cell death.

The anti-tumor activity of mipsagargin has been evaluated in a number of murine models of human tumors, including the MCF-7 human breast tumor model, the PSMA-producing CWR22R-H human prostate tumor xenograft, the SN12C renal cancer xenograft, and the TSU-Pr1 human bladder cancer xenograft, over a range of doses and schedules. Human tumor models include human breast tumor xenografts, human renal tumor xenografts, human bladder tumor xenografts, and human prostate tumor xenografts [20]. Based on these studies, a dose and schedule of mipsagargin administration was identified at which the drug product appeared to be active but without prohibitive toxicity. Subsequently, a Phase I clinical study evaluated the safety of this dose and schedule and determined that the maximum tolerated dose of mipsagargin was 66.8 mg/m^2^ administered by one-hour intravenous infusion on Days 1, 2 and 3 of a 28-day cycle in patients with advanced solid tumors [26].

For this Phase II study, the screening phase was planned to test two dose levels starting at 40 mg/m^2^ on Days 1–3 and increasing to reach the maximum tolerated dose. However, one patient experienced dose-limiting toxicity after dosing reached 66.8 mg/m^2^ on Days 2 and 3, therefore, the 40 mg/m^2^ dose for Days 1–3 was selected for use during the expansion portion based on investigator preference in this patient population with compromised liver function. In total, 19 patients were enrolled at a dose level of 40 mg/m^2^ on Days 1, 2 and 3, and six patients at 40 mg/m^2^ on Day 1 and 66.8 mg/m^2^ on Days 2 and 3. Plasma concentrations of mipsagargin measured at the end of infusion on Day 1 and at pre-dose on Day 2 and Day 3 were within the range of plasma concentrations predicted from the previous Phase I study of mipsagargin in patients with advanced solid tumors [26]. This suggests that the pharmacokinetic profile is consistent in cirrhotic patients with advanced HCC with either compensated or compromised liver function.

Efficacy data were available for 19 patients. No patient experienced CR or PR, which is consistent with other reports of only modest or minimal objective response rates amongst agents that target tissue vasculature, including angiogenic agents such as sorafenib [3,4]. SD was observed in 12 of 19 patients, representing a disease control rate of 63.2% in patients who had progressed on prior treatment with sorafenib. The primary efficacy endpoint was TTP after treatment with mipsagargin in patients with progressive, advanced HCC. When censored data were included, patients in this study had a median TTP of 134.0 days (95% CI: 58.0, NE), which was significantly greater (*p* < 0.001) than that of the historic median TTP of approximately 63 days [28,29,30]. Study patients also had a median PFS of 129.0 days and a median OS of 205.0 days. Sensitivity analyses supported findings for each endpoint.

Guidelines for the use of DCE-MRI in early-phase clinical trials have been proposed [41,42]. One of the most widely recommended endpoints to evaluate vascular response is K_trans_. Many early-phase clinical trials of anti-angiogenic therapy have shown a significant reduction of K_trans_, suggesting the reduction of vascular permeability. Correlations of DCE-MRI parameters with treatment response and other clinical outcomes have been actively explored in many different cancers. K_trans_ measured by DCE-MRI correlated well with tumor response and survival in HCC patients who received sorafenib plus metronomic tegafur/uracil therapy [43]. Five patients, collectively with 11 HCC lesions, had assessable DCE-MRI scans both before initiation of treatment and within three days of receiving the second cycle of treatment. Among these five patients and in all 11 lesions, K_trans_ was reduced; across the 11 lesions, the reduction in K_trans_ averaged 52% (range 13–90%). Changes in contrast enhancement after treatment suggest a clinical benefit of treatment with mipsagargin.

Immunohistochemical evaluation of PSMA expression in archival tumor tissue specimens for 13 patients for whom archival tissue was available found no clear relationship or correlation between PSMA expression and any clinical outcome, safety observation, or other study parameter. Therefore, PSMA expression, at present, should not be considered as a suitable biomarker for response and further evaluation would be required to establish the suitability of PSMA as a potential biomarker for response. Further, no trends were detected in changes in the AFP concentration while on therapy, which is expected given that no objective responses were noted in this study.

In terms of safety, our study permitted Child-Pugh B patients, and despite seven out of 25 patients having Child-Pugh B7 and one patient having Child-Pugh B8, mipsagargin was relatively well tolerated. Only one patient experienced dose-related toxicity in the safety run-in portion of the study (Grade 4 thrombocytopenia, Grade 3 creatinine increase and renal injury). This led to study expansion at the modified dosing of 40 mg/m^2^ day 1–3. Five patients had AEs that led to the discontinuation of treatment; however, no AEs resulted in death. All study patients experienced AEs considered possibly, probably, or definitely related to mipsagargin. However, the majority of these were Grade 1 and 2.

## 5. Conclusions

In conclusion, mipsagargin is a first-in-class PSMA-targeted prodrug that may be an effective therapeutic strategy in patients with HCC, a disease characterized as highly vascularized. Infusion of 40 mg/m^2^ mipsagargin over a 1-hour period on Days 1, 2 and 3 of 28-day cycles was relatively well tolerated and resulted in disease stabilization, decreased tumor blood flow as observed by decrease Ktrans, and prolonged TTP in a population of patients who had progressed on prior treatment with sorafenib. These observations importantly suggest that mipsagargin may have clinical activity in HCC, including in the population of patients with advanced, refractory HCC. In this study, exploratory analyses suggested that mipsagargin decreased blood flow in HCC lesions and in metastatic sites relative to baseline examination within two cycles of exposure to mipsagargin. These findings warrant a larger clinical study to further characterize the activity of mipsagargin in advanced HCC.

## Figures and Tables

**Figure 1 cancers-11-00833-f001:**
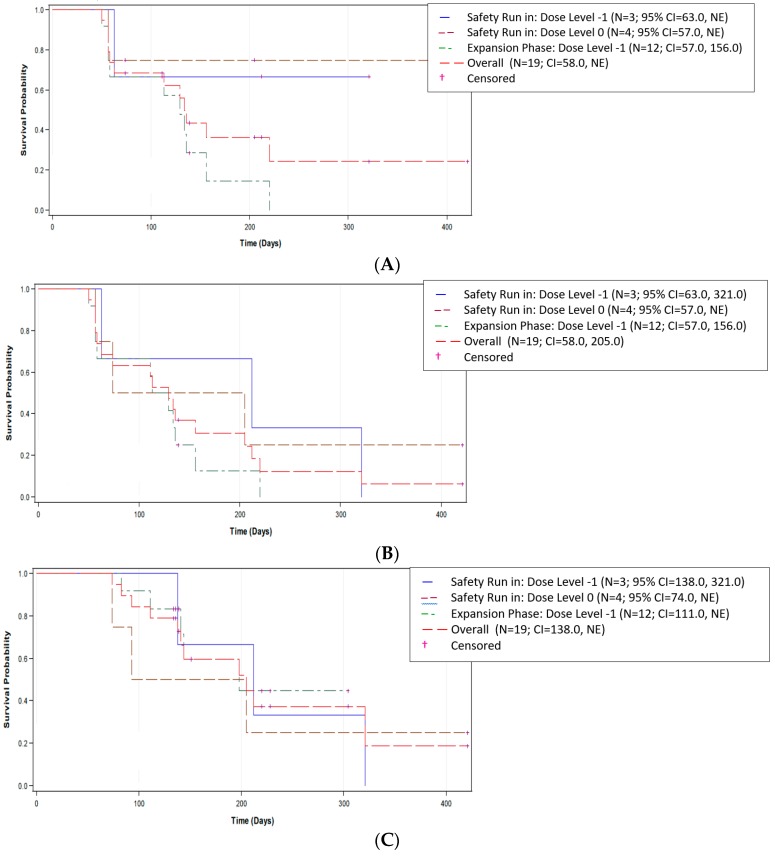
Plot of Kaplan–Meier estimates in the EE population for (**A**) time to disease progression (TTP), (**B**) progression-free survival (PFS), and (**C**) overall survival (OS). TTP was defined as the duration of time from the first dose of mipsagargin to the time of radiologic progression. The progression-free survival (PFS) and overall survival (OS) were estimated using Kaplan–Meier product limit estimates. PFS was defined as the duration of time from the date of the first dose of mipsagargin to the date of radiologic progression or death, whichever occurred first. OS is defined as the duration of time from the date of first dose of mipsagargin to the date of death from any cause.

**Figure 2 cancers-11-00833-f002:**
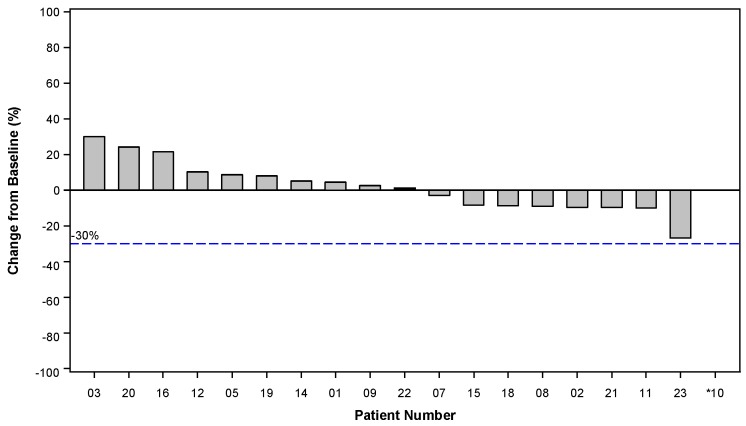
Percentage change in target lesion measurements from baseline. Best Response of the Sum of the Longest Target Lesion Diameters. Patient 10 exhibited new metastatic disease and was removed from the study without measurement of target lesions.

**Figure 3 cancers-11-00833-f003:**
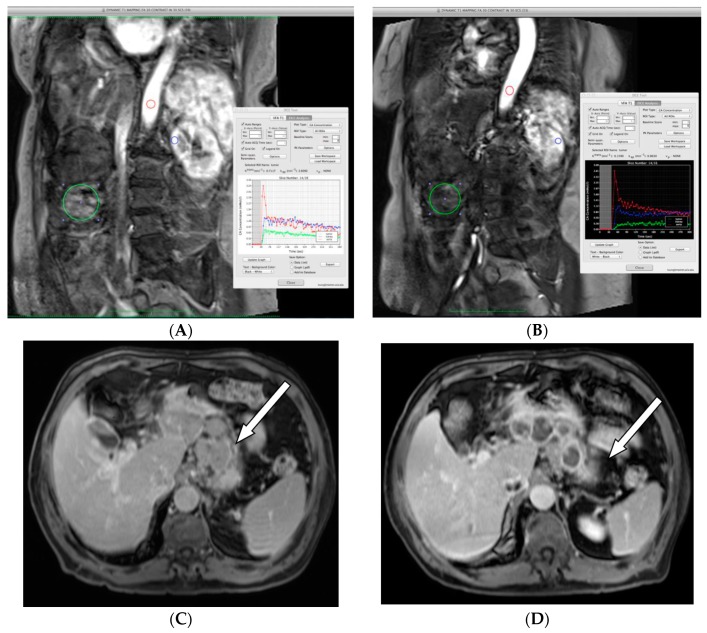
Effect of mipsagargin on blood flow metrics in hepatocellular carcinoma (HCC; **A** and **B**) and in a metastatic lymph node in HCC (**C** and **D**). (**A**,**B**) Radiograph of a patient with multifocal disease that included a previously treated large lesion in the inferior right hepatic lobe. Directly inferior and lateral to the large treated lesion, a 5 cm rounded lesion with arterial phase hyper-enhancement and prompt washout on the baseline contrast-enhanced magnetic resonance imaging (DCE-MRI) exam was selected for analysis. (**A**) DCE-MRI at baseline measured K_trans_ = 0.72 − 0.76 min^−1^. (**B**) DCE-MRI following C2 and measured K_trans_ = 0.14 − 0.16 min^−1^. (Red = aorta; blue = kidney; green = tumor. Range reflects use of larger and smaller regions of interest (ROIs) within the lesion. (**C**,**D**) Gastrohepatic metastatic lymph node involvement, indicated by the arrow, in a patient with advanced HCC. (**C**) DCE-MRI imaging assessment before treatment with mipsagargin. (**D**) Increased hypoenhancement after treatment with mipsagargin suggests response to treatment.

**Table 1 cancers-11-00833-t001:** Patient baseline demographics and clinical characteristics.

Parameter	Safety Run-in Phase			Expansion Phase	Overall (*n* = 25)
	40 mg/m^2^ Days 1–3 (*n* = 3)	40 mg/m2 Day 1 + 66.8 mg/m^2^ Days 2, 3 (*n* = 6)	Total Run-in (*n* = 9)	40 mg Days 1–3 (*n* = 16)	
Median age, y (range)	64.0 (63, 72)	62.5 (53, 69)	64.0 (53, 72)	65.5 (51, 73)	64.0 (51, 73)
*n* (%) ≤ 65 years	2 (66.7%)	4 (66.7%)	6 (66.7%)	8 (50.0%)	14 (56.0%)
>65 years	1 (33.3%)	2 (33.3%)	3 (33.3%)	8 (50.0%)	11 (44.0%)
Female:Male ratio	0:3	2:4	2:7	5:11	7:18
**Race (*n* (%))**					
Asian	0	0	0	1 (6.3%)	1 (4.0%)
Black/African American	0	0	0	2 (12.5%)	2 (8.0%)
White	1 (33.3%)	5 (83.3%)	6 (66.7%)	10 (62.5%)	16 (64.0%)
Other	2 (66.7%)	1 (16.7%)	3 (33.3%)	3 (18.8%)	6 (24.0%)
**ECOG performance status (*n* (%))**					
0	1 (33.3%)	2 (33.3%)	3 (33.3%)	5 (31.3%)	8 (32.0%)
1	2 (66.7%)	4 (66.7%)	6 (66.7%)	11 (68.8%)	17 (68.0%)
**Child-Pugh-score (*n* (%))**					
A5	1 (33.3%)	4 (66.7%)	5 (55.6%)	5 (31.3%)	10 (40.0%)
A6	1 (33.3%)	2 (33.3%)	3 (33.3%)	5 (31.3%)	8 (32.0%)
B7	1 (33.3%)	0	1 (11.1%)	5 (31.3%)	6 (24.0%)
B8	0	0	0	1 (6.3%)	1 (4.0%)
**Best response to previous sorafenib therapy (*n* (%))**					
CR	0	0	0	0	0
PR	0	0	0	0	0
SD	2 (66.7%)	4 (66.7%)	6 (66.7%)	9 (56.3%)	15 (60.0%)
PD	1 (33.3%)	2 (33.3%)	3 (33.3%)	5 (31.3%)	8 (32.0%)
Unknown	0	0	0	2 (12.5%)	2 (8.0%)

Abbreviation: CR, complete response; PD, progressive disease; PR, partial response; SD, stable disease.

**Table 2 cancers-11-00833-t002:** Events occurring in ≥20% of patients in descending order of frequency.

Adverse Event (N [%])	Safety Run-in Phase			Expansion Phase	Overall (*n* = 25)
	40 mg Days 1–3 (*n* = 3)	40 mg Day 1 + 66.8 mg Days 2–3 (*n* = 6)	Total in Safety Run-in (*n* = 9)	40 mg Days 1–3 (*n* = 16)	
Patients with any AE	3 (100%)	6 (100%)	9 (100%)	16 (100%)	25 (100%)
Blood creatinine increased	2 (66.7%)	5 (83.3%)	7 (77.8%)	10 (62.5%)	17 (68.0%)
Fatigue	1 (33.3%)	3 (50.0%)	4 (44.4%)	10 (62.5%)	14 (56.0%)
ALT increased	2 (66.7%)	4 (66.7%)	6 (66.7%)	5 (31.3%)	11 (44.0%)
Nausea	1 (33.3%)	3 (50.0%)	4 (44.4%)	7 (43.8%)	11 (44.0%)
AST increased	1 (33.3%)	3 (50.0%)	4 (44.4%)	6 (37.5%)	10 (40.0%)
Blood bilirubin increased	2 (66.7%)	0	2 (22.2%)	8 (50.0%)	10 (40.0%)
Decreased appetite	2 (66.7%)	2 (33.3%)	4 (44.4%)	6 (37.5%)	10 (40.0%)
Pruritus	1 (33.3%)	2 (33.3%)	3 (33.3%)	7 (43.8%)	10 (40.0%)
Diarrhoea	2 (66.7%)	2 (33.3%)	4 (44.4%)	5 (31.3%)	9 (36.0%)
Hyperbilirubinaemia	0	3 (50.0%)	3 (33.3%)	5 (31.3%)	8 (32.0%)
Rash	1 (33.3%)	1 (16.7%)	2 (22.2%)	6 (37.5%)	8 (32.0%)
Thrombocytopenia	0	3 (50.0%)	3 (33.3%)	5 (31.3%)	8 (32.0%)
Blood ALP increased	2 (66.7%)	1 (16.7%)	3 (33.3%)	4 (25.0%)	7 (28.0%)
Ascites	1 (33.3%)	1 (16.7%)	2 (22.2%)	4 (25.0%)	6 (24.0%)
Blood LDH increased	1 (33.3%)	0	1 (11.1%)	5 (31.3%)	6 (24.0%)
Blood urea increased	2 (66.7%)	2 (33.3%)	4 (44.4%)	2 (12.5%)	6 (24.0%)
Hiccups	1 (33.3%)	1 (16.7%)	2 (22.2%)	4 (25.0%)	6 (24.0%)
Oedema peripheral	2 (66.7%)	3 (50.0%)	5 (55.6%)	1 (6.3%)	6 (24.0%)
Pain in extremity	1 (33.3%)	1 (16.7%)	2 (22.2%)	4 (25.0%)	6 (24.0%)
Vomiting	1 (33.3%)	3 (50.0%)	4 (44.4%)	2 (12.5%)	6 (24.0%)
Anaemia	1 (33.3%)	2 (33.3%)	3 (33.3%)	2 (12.5%)	5 (20.0%)
Tachycardia	2 (66.7%)	1 (16.7%)	3 (33.3%)	2 (12.5%)	5 (20.0%)
Blood phosphorus increased	1 (33.3%)	2 (33.3%)	3 (33.3%)	2 (12.5%)	5 (20.0%)
Blood potassium increased	0	2 (33.3%)	2 (22.2%)	3 (18.8%)	5 (20.0%)
Weight decreased	2 (66.7%)	0	2 (22.2%)	3 (18.8%)	5 (20.0%)
Hyperglycaemia	1 (33.3%)	1 (16.7%)	2 (22.2%)	3 (18.8%)	5 (20.0%)
Hypokalaemia	0	1 (16.7%)	1 (11.1%)	4 (25.0%)	5 (20.0%)
Hepatic encephalopathy	0	0	0	5 (31.3%)	5 (20.0%)

Abbreviations: AE, adverse event; ALP, alkaline phosphatase; ALT, alanine aminotransferase; AST, aspartate aminotransferase; LDH, lactate dehydrogenase; PT, preferred term. Note: If a patient experienced multiple episodes of the same event, the patient was only counted once for that particular event.

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
