# Peer review of "A Phase II, Multicenter, Single-Arm Study of Mipsagargin (G-202) as a Second-Line Therapy Following Sorafenib for Adult Patients with Progressive Advanced Hepatocellular Carcinoma"

_cancers, 2019, doi:10.3390/cancers11060833_

Round 1
Reviewer 1 Report
To improve the manuscript, I suggest to put data on the length of previous sorafenib treatment - if possible. This could create a comparison between the ineffectiveness of treatment with each drug in enrolled HCC patients.
Please correct the editorial errorsmarked in the text.
Author Response
To improve the manuscript, I suggest to put data on the length of previous sorafenib treatment - if possible. This could create a comparison between the ineffectiveness of treatment with each drug in enrolled HCC patients.
The reviewer brings up a point that is of growing relevance given recent approvals of second line therapy options. Patients who may have had longer duration of benefit with sorafenib in the first line, is expected to do better in the second line with further TKI-based therapy. Given our study was started and enrolled prior to approved second line therapies and our study did include some patients beyond 2 lines of therapy as long as they had progressed on sorafenib, the only information we captured on prior of Sorafenib was its best response. The best response to previous sorafenib therapy was SD in 15 patients (60.0%). Eight patients (32.0%) had PD. One patient (4.0%) had an unknown response and 1 patient (4.0%) was classified as not applicable. Unfortunately duration of previous sorafenib was not captured. Of relevance, although all patients has sorafenib, a further 3 patients (12.0%) had received doxorubicin and 3 patients (12%) had received other chemotherapeutics in the past. This has been added into text for clarification.
Reviewer 2 Report
1. About the PSMA, since there are several variants, which one is specific highly expressed in HCC?
2. Some places should better double check to make the writing correctly?
3. About the table 1, the patients baseline, how about the patient risk factors, such as virus infection or not, alcohol or high-fat diet involved in the disease? Usually, the ratio of male to female in HCC is 2:1, while the study showed female /male HCC ratio is about 2:1, could the author give some discussion in the final part?
4. Figure 1 is not clear, could the author provide good resolution pictures? And add some detail description in the figure legend part.
5. Did the author analyze the correlation the percentage changes with other blood test results, and which test could be used for treatment response evaluation? And what’s the reason for the different response to a similar treatment, so what can we learn from the clinical trial study?
About Pharmacodynamic Assessments part, the author showed there is no relationship between clinic response and PSMA, since Mipsagargin (G-202) kindly target to PSMA, did the author test different PSMA, and is there any other target by Mipsagargin? Did the author test the blood flow metric with the treatment time course in a short time and long term, so could the author put more time point MRI figures of typical patients before and after treatment( in supplementary part)?
Author Response
1. About the PSMA, since there are several variants, which one is specific highly expressed in HCC?
Prostate specific membrane antigen (PSMA) is a folate gamma glutamyl carboxypeptidase that is oriented on the plasma membrane of normal and prostate cancer cells. A cytosolic version of PSMA, PSM', results from alternative splicing of the PSMA gene. Two additional alternatively spliced variants of PSMA, PSM-C and PSM-D, have been described as the reviewer has rightly pointed out. Analysis of these works were based on mRNA analysis. The majority of the published studies on PSMA protein levels in patient's tissues have used immunohistochemical analysis. In recent years, researchers have found that PSMA is also selectively expressed in tumor-associated vasculature in a variety of solid tumors. In our study we used the known high expression of PSMA in HCC tumor vasculature to justify the study with Mipsagargin. There are no data that we are aware of that has assessed PSMA or its variants in HCC. We have added test to the paper regarding this variants.
2. Some places should better double check to make the writing correctly?
Grammar checks done and edited in paper.
3. About the table 1, the patients baseline, how about the patient risk factors, such as virus infection or not, alcohol or high-fat diet involved in the disease? Usually, the ratio of male to female in HCC is 2:1, while the study showed female /male HCC ratio is about 2:1, could the author give some discussion in the final part?
Thank you for pointing that discrepancy. The error was the labeling of male:female ratio on the table. The study actually enrolled more male patients (N = 18 [72.0%]) than female patients (N = 7 [28.0%]) – which is expected given the patient population. This has been corrected in the table and a line of that added to results section on the section of patients.
4. Figure 1 is not clear, could the author provide good resolution pictures? And add some detail description in the figure legend part.
A new figure with better resolution added and description of figure legend expanded.
5. Did the author analyze the correlation the percentage changes with other blood test results, and which test could be used for treatment response evaluation? And what’s the reason for the different response to a similar treatment, so what can we learn from the clinical trial study?
Trends in changes in AFP concentration were not detected with therapy. This was the only blood based correlation done. Language on this has been added to discussion.
6. About Pharmacodynamic Assessments part, the author showed there is no relationship between clinic response and PSMA, since Mipsagargin (G-202) kindly target to PSMA, did the author test different PSMA, and is there any other target by Mipsagargin? Did the author test the blood flow metric with the treatment time course in a short time and long term, so could the author put more time point MRI figures of typical patients before and after treatment( in supplementary part)?
Again in relevance to point 1 above, the target for Mipsagargin is the PSMA. We have not tested its effect on the other PSM variants and certainly it’s something we need to do pre-clinically including evaluating the PSMA and its variants in HCC vasculature.
For patients who had a DCE-MRI scan at baseline, the follow-up scan was performed within 72 hours after infusion of G-202 on Cycle 2, Day 3. The volume transfer coefficient, Ktrans, was calculated based using the arterial input function derived from the signal in the abdominal aorta, using a standard Tofts model for Ktrans calculation. The range of baseline measured Ktrans = 0.72 0.76 min-1 and following C2 measured Ktrans = 0.14-0.16 min-1 and reported for the 5 patients with available paired analysis. The figures shown were representation at the measured location for these values. No longer time points of measurements were done or available.
Reviewer 3 Report
- Patient ethnicity is relevant for this patient group (Li J et al. Factors associated with ethnical disparity in overall survival for patients with hepatocellular carcinoma. Oncotarget. 2017 Feb 28;8(9):15193-15204. doi: 10.18632/oncotarget.14771). Is ethnicity known for these patients? Was ethnicity related to response?
- Background of the cancer is important, especially cirrhosis (van Meer et al. Hepatocellular carcinoma in cirrhotic versus noncirrhotic livers: results from a large cohort in the Netherlands. Eur J Gastroenterol Hepatol. 2016 Mar;28(3):352-9). Were results different in patients with and without accompanying cirrhosis?
.
Author Response
Patient ethnicity is relevant for this patient group (Li J et al. Factors associated with ethnical disparity in overall survival for patients with hepatocellular carcinoma. Oncotarget. 2017 Feb 28;8(9):15193-15204. doi: 10.18632/oncotarget.14771). Is ethnicity known for these patients? Was ethnicity related to response?
- Background of the cancer is important, especially cirrhosis (van Meer et al. Hepatocellular carcinoma in cirrhotic versus noncirrhotic livers: results from a large cohort in the Netherlands. Eur J Gastroenterol Hepatol. 2016 Mar;28(3):352-9). Were results different in patients with and without accompanying cirrhosis?
The reviewer brings up a good point in terms of clinical efficacy based on race/ethnicity.
Table does list the patients race. In terms of ethnicity, majority of patients (N = 16 [64.0%]) were white and not Hispanic or Latino. We have added this into text in the results section on patient demographics. Unfortunately correlation with ethnicity and clinical efficacy was not found likely given the small patient numbers.
All patients in this study had clinical evidence of cirrhosis. There were no biopsy confirmation of cirrhosis from liver for most patients apart from the tumor biopsy of HCC on a background of known cirrhosis. In this study hepatitis C as the cause for the cirrhosis was known for 15 patients (60.0%). The rest non-hepatitis related cirrhosis (either alcohol or non-alcoholic steotohepatitis), and language of this has been added to the study text.